# Clinical Implication and Prognostic Value of Artificial-Intelligence-Based Results of Chest Radiographs for Assessing Clinical Outcomes of COVID-19 Patients

**DOI:** 10.3390/diagnostics13122090

**Published:** 2023-06-16

**Authors:** Hyun Joo Shin, Min Hyung Kim, Nak-Hoon Son, Kyunghwa Han, Eun-Kyung Kim, Yong Chan Kim, Yoon Soo Park, Eun Hye Lee, Taeyoung Kyong

**Affiliations:** 1Department of Radiology, Research Institute of Radiological Science and Center for Clinical Imaging Data Science, Yongin Severance Hospital, Yonsei University College of Medicine, Yongin-si 16995, Republic of Korea; lamer-22@yuhs.ac (H.J.S.); ekkim@yuhs.ac (E.-K.K.); 2Center for Digital Health, Yongin Severance Hospital, Yonsei University College of Medicine, Yongin-si 16995, Republic of Korea; 3Division of Infectious Diseases, Department of Internal Medicine, Yongin Severance Hospital, Yonsei University College of Medicine, Yongin-si 16995, Republic of Korea; mhkim16@yuhs.ac (M.H.K.); amomj@yuhs.ac (Y.C.K.); ysparkok2@yuhs.ac (Y.S.P.); 4Department of Statistics, Keimyung University, Daegu 42601, Republic of Korea; nhson@ms.kmu.ac.kr; 5Department of Radiology, Severance Hospital, Research Institute of Radiological Science and Center for Clinical Imaging Data Science, Yonsei University College of Medicine, Seoul 03722, Republic of Korea; khhan@yuhs.ac; 6Division of Pulmonology, Allergy and Critical Care Medicine, Department of Internal Medicine, Yongin Severance Hospital, Yonsei University College of Medicine, Yongin-si 16995, Republic of Korea; hieunhye@yuhs.ac; 7Department of Hospital Medicine, Yongin Severance Hospital, Yonsei University College of Medicine, Yongin-si 16995, Republic of Korea

**Keywords:** artificial intelligence, COVID-19, lung diseases, prognosis, software

## Abstract

This study aimed to investigate the clinical implications and prognostic value of artificial intelligence (AI)-based results for chest radiographs (CXR) in coronavirus disease 2019 (COVID-19) patients. Patients who were admitted due to COVID-19 from September 2021 to March 2022 were retrospectively included. A commercial AI-based software was used to assess CXR data for consolidation and pleural effusion scores. Clinical data, including laboratory results, were analyzed for possible prognostic factors. Total O_2_ supply period, the last SpO_2_ result, and deterioration were evaluated as prognostic indicators of treatment outcome. Generalized linear mixed model and regression tests were used to examine the prognostic value of CXR results. Among a total of 228 patients (mean 59.9 ± 18.8 years old), consolidation scores had a significant association with erythrocyte sedimentation rate and C-reactive protein changes, and initial consolidation scores were associated with the last SpO_2_ result (estimate −0.018, *p* = 0.024). All consolidation scores during admission showed significant association with the total O_2_ supply period and the last SpO_2_ result. Early changing degree of consolidation score showed an association with deterioration (odds ratio 1.017, 95% confidence interval 1.005–1.03). In conclusion, AI-based CXR results for consolidation have potential prognostic value for predicting treatment outcomes in COVID-19 patients.

## 1. Introduction

The coronavirus disease 2019 (COVID-19) caused by the severe acute respiratory syndrome coronavirus 2 (SARS-CoV-2) has become a global pandemic disease since 2019 [1]. Compared with two other coronavirus-related diseases in the past two decades, COVID-19 is more widespread and has a fatality rate of 1.4% [2]. This virus is easily spread through respiratory droplets and causes frequent radiologic and laboratory abnormalities after exposure. Even with massive global efforts, the disease has not been completely conquered yet because of the wide genetic recombination and variations of the virus, which causes frequent hospitalization, mortalities and adverse outcomes [1].

Artificial intelligence (AI) is being widely applied in medicine, and efforts have focused on integrating AI for the efficient diagnosis, treatment, and prediction of outcomes of COVID-19 patients [3,4,5]. COVID-19 manifests radiologic abnormalities on chest radiographs (CXR), such as peripherally located increased opacity areas with lower lobe predominance, even though these findings overlap with those of other diseases, such as organizing pneumonia or other infectious diseases [6,7]. These CXR abnormalities were used to develop an AI algorithm to detect COVID-19 infection, and recently some researchers attempted to demonstrate the prognostic implications of detecting CXR abnormalities using AI [3,8,9,10]. Previous researchers developed their own algorithm for predicting mortality from COVID-19 by analyzing CXR data in addition to the clinical data [8,10,11]. They demonstrated that their respective algorithms had good prognostic performance and showed that the addition of an AI algorithm for CXR to existing prognostic factors improved performance. However, there is still a lack of studies demonstrating whether AI-based CXR results could reflect the clinical course and show prognostic value during the treatment of COVID-19 infection.

Therefore, the purpose of this study was to investigate the clinical implication and prognostic value of AI-based CXR results in addition to the clinical indicators for assessing the disease course and outcome of COVID-19 infection.

## 2. Materials and Methods

The Institutional Review Board (IRB) of Yongin Severance Hospital approved this retrospective study (IRB number 9-2022-0046) and the requirement of informed consent was waived due to the study’s retrospective design. The study was conducted according to the guidelines of the Declaration of Helsinki and Strengthening the Reporting of Observational Studies in Epidemiology (STROBE).

### 2.1. Subjects and Clinical Features

Patients who were admitted to our hospital due to COVID-19 infection from September 2021 to March 2022 were retrospectively included. Inclusion criteria were as follows: (1) positive result of COVID-19 PCR and admission to our hospital during the study period, and (2) CXR exams and laboratory results more than once during the admission dates. The exclusion criteria were as follows: (1) patients under 18 years of age at the time of diagnosis of COVID-19 infection, and (2) patients who did not undergo CXR or laboratory results during the admission.

After reviewing electronic medical records, age, sex, and laboratory results were recorded. The laboratory results, including WBC (103/μL), neutrophil (%), lymphocyte (%), erythrocyte sedimentation rate (ESR, mm/h), C-reactive protein (CRP, mg/L), prothrombin time (PT, s, %), aspartate aminotransferase (AST, IU/L), and alanine aminotransferase (ALT, IU/L) results, known to be associated with severity of COVID-19 infection during the admission were evaluated [4]. If the patients had repeated laboratory exams, the repeated results within 3 days of all CXRs taken during admission dates were all recorded.

To evaluate outcome after treatment, clinical features such as total O_2_ application period (days), last SpO_2_ (%), and deterioration at the discharge date were evaluated. Because our hospital is a general hospital, after the treatment of COVID-19 infection following the National Institutes of Health guidelines [12], patients who needed intensive treatment including ventilator care and invasive procedures for COVID-19 infection were transferred according to the national guidelines. Therefore, if the patients were getting worse without recovery, they were discharged and transferred to hospitals dedicated to COVID-19 treatment; others who had recovered well during the admission without need for O_2_ supply and any symptoms were discharged without transferring. Therefore, we could assess whether the patient deteriorated or not at the time of discharge and used this information as one of the prognostic results. The total O_2_ application period was defined as the days needed to supply O_2_ during the admission because the patient could not sustain O_2_ saturation in room air. The last SpO_2_ was SpO_2_ at the time of discharge date and this was used as one of the prognostic factors of treatment outcomes.

### 2.2. AI-Based CXR Results

As an imaging manifestation of COVID-19 infection, increased opacity area and pleural effusion on CXR were evaluated using AI-based lesion-detection software. COVID-19 infection exhibits increased opacity areas, such as ground-glass opacity (GGO) and consolidation, mainly in subpleural spaces of both lungs and is rarely associated with pleural effusion. In our hospital, commercially available AI-based lesion-detection software (Lunit INSIGHT CXR, version 3, Lunit Inc., South Korea) was integrated into all CXRs in the pictures archiving and communication system (PACS). This software could detect eight lesions on CXR, including consolidation (increased opacity area on CXR, encompassing the terms of GGO and consolidation) and pleural effusion [13,14]. In the previous study, the diagnostic accuracy for consolidation and pleural effusion using the same algorithm was 93% and 99.2%, respectively [15]. When considering consolidation, nodules, and pneumothorax together, the AI algorithm’s stand-alone performance had an area-under-the-curve (AUC) of 0.867 with a sensitivity of 88.5% and specificity of 72.3% [16]. To determine the presence of each lesion, the software presents an abnormality score ranging from 0 to 100%; this score represents the probability that a given CXR has lesions as determined by AI. When the abnormality score of each lesion type is above 15% [17], this software determines that a lesion is present on the CXR and displays a contour map and abbreviation (where “Csn” stands for consolidation and “Pef” stands for pleural effusion, as pre-defined format in the software) with the abnormality score for each lesion (Figure 1). For the study population, we extracted abnormality scores of consolidation and pleural effusion for CXRs that the patients underwent during the admission dates. For patients with repeated exams, all results of each follow-up day were extracted from the server.

### 2.3. Statistical Analysis

Data were analyzed by SAS 9.4 (SAS Institute, Cary, NC, USA) and SPSS (version 26.0, IBM Corp, Armonk, NY, USA). After the normality test using the Shapiro–Wilk test, continuous values were presented as means and standard deviations. To determine the association between patterns of clinical and CXR results during the admission period, a generalized linear mixed model was used.

To examine the prognostic value of the initial CXR and laboratory results for the prediction of total O_2_ supply period, last SpO_2_, and deterioration after the treatment, univariate and multivariate linear regression and logistic regression were used. To evaluate whether the pattern of change in all CXR and laboratory results during the admission period could be used to predict the prognostic outcome, the generalized linear mixed model was used for only patients who have more than two repeated exams. We also evaluated whether early changes in CXR and laboratory results during the first 3 days could be used to predict the prognosis of COVID-19 infection using linear regression and logistic regression tests. In this retrospective study, the timing of the first follow-up CXR after hospitalization can vary among patients. To account for the magnitude of early changes and evaluate treatment response, the approach of calculating the gradient of sequentially measured values was used. Therefore, early changes of results in the first 3 days were calculated using an equation. The formula for calculating the changing degree in the first 3 days is as follows: (measured results of the second follow-up exam minus the results of the first exam at admission) divided by the follow-up days between the second and first exams, multiplied by 3 days. This was also evaluated for patients who had repeated CXR or laboratory results. A *p*-value less than 0.05 was regarded as statistically significant.

## 3. Results

### 3.1. Patients and Clinical Results

During the study period, a total of 228 patients (Male: Female = 135:93, mean age 59.9 ± 18.8 years old) who had positive results for COVID-19 reverse transcriptase polymerase chain reaction (RT-PCR) and were admitted to our hospital were included in this study. During the study period, from the third week of January 2022, the Omicron variant became predominant in South Korea, while the Delta variant was dominant prior to that period. No one was excluded due to lack of CXR or laboratory results during the admission period. Among the 228 patients, 136 patients had repeated CXR and laboratory exams (mean 1.9 times, maximum 8 times). The mean admission period was 9 days, ranging from 1 to 33 days. Approximately 22.4% (51/228) of patients were asymptomatic. Demographics of the patients and exams are summarized in Table 1. The mean abnormality score for consolidation was 36.6 ± 33.4% for the initial 228 CXRs and 49.7 ± 34.8% for all 463 CXRs including repeated exams. The mean abnormality score for pleural effusion was 3.2 ± 10% for the initial CXR and 3.4 ± 8.8% for all CXRs. As a prognostic outcome, total O_2_ supply period was 2.6 ± 18.8 days and the last SpO_2_ was 94.7 ± 2.6%. Approximately 11.8% (27/228) patients showed deterioration and were transferred to dedicated hospitals for intensive care.

### 3.2. Association between Pattern of Change in Clinical and CXR Results for COVID-19 Patients

Table 2 shows the results of association between the pattern of change in clinical and CXR results during the admission period. In univariate analysis, patterns of neutrophil, lymphocyte, ESR, CRP and AST results showed a significant association with the change in consolidation score on CXR in each patient (all, *p* < 0.001). In multivariate analysis, the pattern of change in ESR and CRP showed a significant association with consolidation score (*p* ≤ 0.001). For pleural effusion, neutrophil and lymphocytes showed significant association on univariate analysis but no significant result on multivariate analysis.

### 3.3. Prognostic Value of CXR and Laboratory Results for the Prediction of Outcome

Table 3 shows the results for the prognostic value of the initial CXR and laboratory results at the time of admission for predicting prognostic outcomes including total O_2_ supply period, last SpO_2_, and deterioration. In multivariate analysis, the initial WBC result showed significant association with the total O_2_ supply period (estimate 0.005, *p* = 0.022). Initial consolidation score (estimate −0.018, *p* = 0.024), CRP (estimate −0.02, *p* = 0.007), and ALT (estimate −0.037, *p* = 0.023) showed a significant association for the last SpO_2_ result. For the aspect of deterioration, consolidation score, neutrophil, lymphocyte, ESR, and CRP showed significant association in univariate logistic regression analysis but showed no significant association in multivariate analysis.

To determine whether the patterns of results during the admission period showed a significant association with prognostic outcomes, we evaluated 136 patients who had repeated CXR and laboratory exams (Table 4). On multivariable analysis, consolidation score and CRP results showed a significant association with the total O_2_ supply period (*p* = 0.013 and 0.009, respectively) and with the last SpO_2_ (*p* = 0.037 and 0.047, respectively). In addition, CRP level showed a significant association with deterioration (estimate 0.009, *p* = 0.018).

We also examined whether the early changes of CXR and laboratory results during the first 3 days after admission could predict prognostic outcome, and the results are presented in Table 5. Only the changing degree of consolidation score between 3 days after treatment and at admission date showed a significant association with deterioration (odds ratio 1.017, 95% confidence interval 1.005–1.03, *p* = 0.005).

## 4. Discussion

Our results demonstrated that clinical features such as ESR and CRP levels showed a significant association with patterns of change in consolidation on CXR determined by AI in COVID-19 patients. Increased ESR and CRP levels were associated with an increased consolidation score on CXR during the admission (beta = 0.417, 0.231, respectively). We examined the prognostic value of CXR and laboratory results for COVID-19 patients and found that a lower initial consolidation score, as well as CRP and ALT levels, showed a significant association with a higher last SpO_2_ result, while a higher initial WBC result showed a significant association with a longer total O_2_ supply period. In addition to these findings with the initial results, increased consolidation scores on all repeated CXRs during the admission period showed significant association with increased total O_2_ supply period and decreased last SpO_2_ result. In addition, increased CRP results in repeated exams showed a significant association with increased O_2_ supply period and decreased last SpO_2_ and also predicted deterioration. Notably, an early changing degree of consolidation score during the first 3 days after admission had a prognostic value on predicting deterioration of COVID-19 admission.

There have been many recent efforts to use AI for diagnosing and validating prognostic power for management of pulmonary infection. Several studies have already been conducted to find the meaning of AI on CXR in diagnosing pneumonia, evaluating treatment response, or predicting prognosis [18,19,20]. In addition to pneumonia, researchers have focused on developing an AI algorithm to detect COVID-19 using CXR because CXR is the first-line imaging study for diagnosing and guiding treatment options of patients with respiratory symptoms [21,22,23,24]. There have also been attempts to apply AI to the diagnosis of COVID-19 using chest computed tomography (CT) in addition to CXR [25]. Some efforts were made to predict the outcome of COVID-19 patients using radiomics features on CXR or AI-based prediction models using CXR [11,26,27,28]. Similar to our study, Mushtaq et al. used Conformité Européenne (CE)-certified AI software on initial CXRs to predict the mortality of COVID-19 patients and the authors demonstrated that initial CXRs could be used as a predictor of adverse outcome in COVID-19 patients [3]. However, this software could quantify areas in pixels with increased opacities on the CXR representing the extent. Another study by Jiao et al. demonstrated that an AI model trained with CXR and clinical data together showed good prognostic performance compared with clinical data or conventional severity scores for predicting progression of COVID-19 patients [8]. Despite several studies aiming to develop and validate AI algorithms, the quantification of CXR results itself using AI software with guaranteed repeatability and generalizability is challenging. It is important to use the clinical implications of AI-based lesion-detection software for CXR—which is a basic tool for screening and follow-up and can be easily used for evaluating therapeutic responsiveness of lung infection patients, especially those with diseases such as COVID-19—where the use of chest CT is not readily available. Understanding its clinical impact on treatment, not just diagnostic accuracy, is also important, but needs more evidence.

Therefore, our study has a strength in using commercially available AI software that was proven to have high diagnostic performance in previous studies, and this could guarantee the relative repeatability of our results in other centers [14,15,29,30,31]. While various AI algorithms have been developed by research teams, overfitting has been a major obstacle to their actual clinical application. However, this study used a commercially available program, which makes it possible for other institutions to apply the results. Whether the abnormality score presented by AI could be used to represent disease severity or extent was questionable. The use of this score in the same way as area of pixels or opacity of images requires caution because of the unexplainable characteristics of AI [32]. However, our results demonstrated that laboratory results of ESR and CRP levels showed a similar pattern with consolidation score on CXR as determined by AI. Therefore, this score could reflect some degree of disease extent and severity presented on the images. From this result, we suggest that AI would present high scores when the lesion was more obvious with clear opacity and a large extent. Starting from this result, we demonstrated that the consolidation score of AI could be used to predict whether the patients would need a longer O_2_ supply period or have a good prognosis using the longitudinal data, not limited to the initial results.

Using an equation, we demonstrated that early changes of CXR results during the first 3 days after admission showed a significant association with deterioration. While examining patients, we came up with the idea that the pattern of early consolidation changes could be quantified by AI and used to evaluate the patient’s treatment response by measuring the degree of increase or decrease. To test this idea, we proposed an equation. However, due to the limitations of retrospective research, it was not possible to obtain consistent results on a uniform date as the follow-up periods for actual patients varied. As an alternative, we devised a method to evaluate the pattern of changes over the first three days of hospitalization. Nevertheless, due to the limited number of patients who underwent follow-up (only 136 out of the total number of patients), most of the predicted outcomes did not yield significant results, as shown in Table 5. However, as the most critical factor was whether the patient’s condition improved or worsened, the initial pattern of consolidation response showed significant results, indicating some promise. Therefore, we believe that a well-established prospective large-scale study will be needed to validate this idea in the future.

While we also included pleural effusion score, it did not show significant results. COVID-19 infection manifests mainly as consolidation or GGO in the subpleural portion of lower lobes [6]. Pleural effusion is uncommon for COVID-19 infection and considered an atypical imaging pattern, similar to other types of viral pneumonia [6]. This could affect the results of our study that showed that the mean abnormality score for pleural effusion was 3.2–3.4% for initial and all repeated CXR, below the cutoff value of 15%. For the 228 initial CXRs, the score range for pleural effusion was 0.12% to 94.42%. Among them, 220 CXRs had a pleural effusion score of less than 15%, resulting in a higher standard deviation. We included pleural effusion in the imaging findings because it is known to be accompanied by consolidation in typical pneumonia. However, this result could be seen as reflecting the fact that COVID-19 pneumonia differs from previous bacterial pneumonia in that it is less likely to be accompanied by pleural effusion, while also serving as evidence that AI results reflect differences in imaging findings due to the actual etiology. Our results showed that the consolidation score of AI reflected well the radiologic characteristics of COVID-19 infection and could be used to determine treatment response as well as the usual visual diagnosis of CXR in routine practice, in addition to the other well-known clinical laboratory results.

During the study period, the Delta and Omicron variants were predominant. There may be concerns about whether the results of this study can be applied uniformly across different variants due to the differences in the strains. The transmission, symptoms, and vaccine effectiveness could be different between two variants. Clinical outcomes suggest that the proportion of severe cases was higher with Delta than with Omicron infections. However, regardless of the variant, the transition to severe cases can occur equally in elderly patients, patients with underlying diseases, and patients with compromised immune systems. Moreover, no differences in imaging findings have been reported between different variants. Therefore, it is expected that the results of this study can be clinically applicable regardless of the variant, as prognosis prediction for severe patients is more important than management for mild patients.

There are several limitations in this study. First, we only used one commercially available AI software. While this could enhance the repeatability compared with other studies with their own AI algorithm, the problem of generalization still exists. The issue of generalization is presented as a limitation in various AI studies. However, to overcome this, we used a commercialized program that has a certain level of performance and diagnostic accuracy, backed by various studies and evidence, for this study [13,14,15,30,31,33]. It is currently applied to all CXR at the hospital of the actual researchers and is used for patient clinical treatment. Although using multiple AI programs together and comparing their performance might be a good method to overcome generalization, it is also a significant research topic and does not align with the purpose of our study. Hence, we aimed to report its clinical significance using a program proven through numerous studies. Therefore, we believe that comparing the performance of various AI programs in future research, as proposed in this study, could be another good research topic, and we hope that researchers will pursue it following this study. Second, the number of the patients with repeated exams was relatively small. Because our hospital was not a dedicated hospital for intensive care of critical COVID-19 patients, disease outcome was limited as the O_2_ supply period, the last SpO_2_, and deteriorated patients who need a transfer, and we could not include mortalities. Third, in this retrospective study, due to slight variations in the timing of follow-up CXR after hospitalization, we could only represent the magnitude of numerical changes using the formula to calculate the gradient. While the presented results were reviewed by expert statisticians, we acknowledge the limitation that it may not fully capture the actual measured values. Therefore, starting from this initial study, further prospective research with a larger patient population is needed to demonstrate the utility of artificial intelligence as a tool for evaluating treatment response and predicting prognosis. At last, we were unable to evaluate or upgrade the diagnostic accuracy of the AI software due to the fact that detecting chest lesions was beyond the scope of this study, and we could not modify the performance of commercially available AI software, as it was not developed by the authors. Nonetheless, we were able to use an AI program with diagnostic performance proven by previous studies to overcome this limitation [13,15,16,30,31,34]. Additionally, we strived to present the most objective and accurate analysis in conjunction with two statistical experts for the research methodology and analytical aspect, while following various methodological proofs and advices [35,36]. In the future, conducting a large prospective study focused on the accuracy of AI software would be critical for advancing this area of research.

## 5. Conclusions

This study showed that AI-based CXR results for consolidation showed a similar pattern to changes in ESR and CRP results during the treatment of COVID-19 and exhibit potential prognostic value for predicting treatment outcomes. This study demonstrated the potential of using AI-based results from CXR to predict disease progression, treatment outcomes, and prognosis in an important respiratory disease.

## Figures and Tables

**Figure 1 diagnostics-13-02090-f001:**
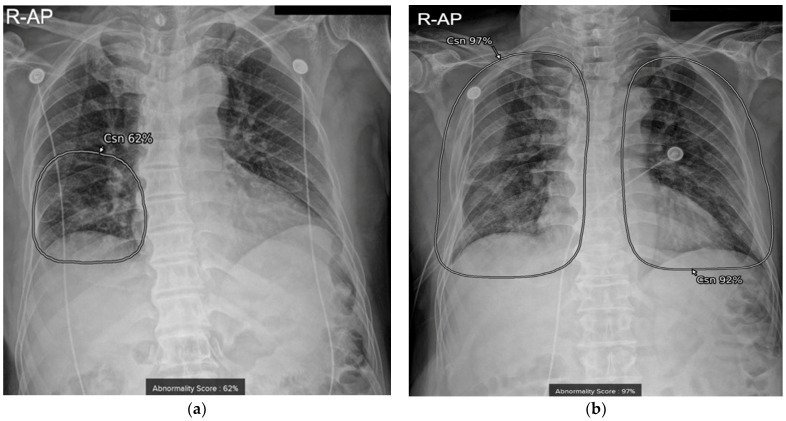
Two chest radiographs (CXR) of a 72-year-old man who had deterioration after treatment for COVID-19. The AI algorithm generates a contour map with abbreviations (where “Csn” stands for consolidation) and its abnormality score for lesion localization. (**a**) Initial CXR taken on the day of admission showed consolidation score of 62% in right lower lung field. His WBC level was 6980/μL, CRP level was 33.3 mg/L, and SpO_2_ was 96% on the same day. (**b**) His follow-up CXR on the 6th day showed increased consolidation score of 97%. His WBC level was 9760/μL, CRP level was 234 mg/L, and SpO_2_ was 90% on the same date. He was transferred to a dedicated hospital for intensive care due to the deterioration.

**Table 1 diagnostics-13-02090-t001:** Results of patients and repeated exams during admission from COVID-19.

	Number of Patients with Initial Exam/with Repeated Exams	Mean ± SD of Patients or Initial Exams	Total Number of Exams Performed during Admission (Min-Max Number of Repeated Exams in a Patient)	Mean ± SD of Total Exams
AI-based abnormality scores on CXR				
Consolidation (%)	228/136	36.6 ± 33.4	463 (1–8)	49.7 ± 34.8
Pleural effusion (%)	228/136	3.2 ± 10.0	463 (1–8)	3.4 ± 8.8
Laboratory results				
WBC	228/136	78.6 ± 570.5	463 (1–8)	42 ± 402
Neutrophil	227/136	63.1 ± 12.6	462 (1–8)	64.3 ± 13.9
lymphocyte	227/136	25.7 ± 10.6	462 (1–8)	25.3 ± 11.6
ESR	204/41	42.3 ± 26.3	276 (1–8)	41.7 ± 26
CRP	220/133	28.4 ± 37.5	453 (1–8)	31.4 ± 44.3
PT (s)	141/14	8.7 ± 1.4	163 (1–5)	8.8 ± 1.8
PT (%)	141/14	121.7 ± 22.4	163 (1–5)	122 ± 23.2
AST	225/116	34.6 ± 25.1	431 (1–8)	35.4 ± 27.1
ALT	223/116	28.8 ± 24.5	431 (1–8)	33.3 ± 29
Prognostic outcomes				
O_2_ supply period (days)	228	2.6 ± 18.8		
Last SpO_2_ (%)	228	94.7 ± 2.6		
Deterioration	27/228 (11.8%)			

Values are presented as mean ± standard deviation or number (percentage or minimum-maximum numbers). Abbreviations: SD = standard deviation, AI = artificial intelligence, CXR = chest radiograph, WBC = white blood cell, ESR = erythrocyte sedimentation rate, CRP = C-reactive protein, PT = prothrombin time, AST = aspartate aminotransferase, ALT = alanine transaminase.

**Table 2 diagnostics-13-02090-t002:** Association between patterns of change in clinical and AI-based CXR results on COVID-19 patients.

AI-Based Abnormality Score on CXR (%)	Consolidation				Pleural Effusion			
	Univariate		Multivariate		Univariate		Multivariate	
	Beta	*p*-Value	Beta	*p*-Value	Beta	*p*-Value	Beta	*p*-Value
WBC	0.003	0.393			0.000	0.962		
Neutrophil	0.768	<0.001	0.551	0.241	0.080	0.040	0.011	0.926
lymphocyte	−0.778	<0.001	0.125	0.816	−0.100	0.032	−0.088	0.537
ESR	0.584	<0.001	0.417	<0.001	0.015	0.456		
CRP	0.239	<0.001	0.231	0.001	0.013	0.257		
PT (s)	−0.816	0.697			−0.071	0.866		
PT (%)	0.016	0.922			−0.009	0.749		
AST	0.201	<0.001	0.036	0.648	0.001	0.972		
ALT	0.069	0.367			−0.013	0.506		

Abbreviations: AI = artificial intelligence, CXR = chest radiograph, WBC = white blood cell, ESR = erythrocyte sedimentation rate, CRP = C-reactive protein, PT = prothrombin time, AST = aspartate aminotransferase, ALT = alanine transaminase.

**Table 3 diagnostics-13-02090-t003:** Prognostic value of initial CXR and laboratory results to predict outcome of COVID-19 patients.

Outcomes (*n* = 228)	Total O_2_ Supply Period (Days)				Last SpO_2_ (%)				Deterioration					
	Univariate		Multivariate		Univariate		Multivariate		Univariate			Multivariate		
	Beta	*p*-Value	Beta	*p*-Value	Beta	*p*-Value	Beta	*p*-Value	OR	95% CI	*p*-Value	OR	95% CI	*p*-Value
AI-based abnormality scores on CXR (%)														
Consolidation	0.129	<0.001	0.053	0.295	−0.035	<0.001	−0.018	0.024	1.031	1.018–1.045	<0.001	1.015	0.998–1.033	0.087
Pleural effusion	0.189	0.129			−0.053	0.019	−0.019	0.342	1.016	0.987–1.045	0.297			
Laboratory results														
WBC	0.007	0.001	0.005	0.022	−0.001	0.014	0.000	0.254	1.000	1.000–1.001	0.058			
Neutrophil	0.329	0.001	0.168	0.605	−0.074	<0.001	0.056	0.266	0.162	1.074–1.182	<0.001	1.076	0.926–1.251	0.339
Lymphocyte	−0.352	0.003	0.023	0.95	0.092	<0.001	0.098	0.095	0.06	0.833–0.934	<0.001	1.006	0.841–1.203	0.946
ESR	0.112	0.031	0.034	0.565	−0.019	0.032	0.006	0.529	0.593	1.008–1.040	0.003	1.003	0.982–1.025	0.778
CRP	0.128	<0.001	0.042	0.369	−0.034	<0.001	−0.02	0.007	1.021	1.012–1.030	<0.001	1.015	0.998–1.033	0.087
PT (s)	0.437	0.76			0.024	0.914			0.927	0.593–1.450	0.74			
PT (%)	−0.076	0.388			0.015	0.261			0.995	0.976–1.014	0.598			
AST					−0.037	<0.001	0.007	0.669	1.01	0.998–1.022	0.101			
ALT					−0.033	<0.001	−0.37	0.023	1.005	0.992–1.019	0.444			

Abbreviations: AI = artificial intelligence, CXR = chest radiograph, OR = odds radio, CI = confidence interval, WBC = white blood cell, ESR = erythrocyte sedimentation rate, CRP = C-reactive protein, PT = prothrombin time, AST = aspartate aminotransferase, ALT = alanine transaminase.

**Table 4 diagnostics-13-02090-t004:** Prognostic value of patterns of change in all CXR and laboratory results during admission period to predict outcome.

Outcomes (*n* = 136)	Total O_2_ Supply Period (Days)				Last SpO_2_ (%)				Deterioration			
	Univariate		Multivariate		Univariate		Multivariate		Univariate		Multivariate	
	Beta	*p*-Value	Beta	*p*-Value	Beta	*p*-Value	Beta	*p*-Value	Beta	*p*-Value	Beta	*p*-Value
Consolidation	0.007	<0.001	0.005	0.013	−0.027	<0.001	−0.018	0.037	0.017	0.01	0.009	0.215
Pleural effusion	0.015	0.147			−0.046	0.101			0	0.986		
WBC	0	0.964			−0.001	0.137			0.001	0.215		
Neutrophil	0.012	0.035	0.005	0.294	−0.052	0.004	0.005	0.933	0.05	0.002	0.058	0.312
Lymphocyte	−0.013	0.054			0.058	0.006	0.03	0.675	−0.053	0.009	0.045	0.504
CRP	0.005	0.002	0.003	0.009	−0.02	<0.001	−0.012	0.047	0.013	<0.001	0.009	0.018

Abbreviations: CXR = chest radiograph, WBC = white blood cell, CRP = C-reactive protein.

**Table 5 diagnostics-13-02090-t005:** Prognostic value of early changes of CXR and laboratory results during the first 3 days for predicting outcome.

Outcomes (*n* = 136)	Total O_2_ Supply Period (Days)		Last SpO_2_ (%)		Deterioration	
	Univariate		Univariate		Univariate	
	Beta	*p*-Value	Beta	*p*-Value	OR	95% CI
Consolidation	0.001	0.729	−0.001	0.815	1.017 ^a^	1.005–1.03
Pleural effusion	0.012	0.443	0.011	0.839	1.12	0.985–1.274
WBC	0	0.717	0	0.312	1	0.999–1.000
Neutrophil	0.004	0.273	0.001	0.952	0.993	0.966–1.019
Lymphocyte	−0.001	0.851	−0.008	0.578	1.012	0.983–1.042
CRP	0	0.742	−0.002	0.597	1.008	0.998–1.019

^a^ *p*-value = 0.005. Abbreviations: CXR = chest radiograph, OR = odds radio, CI = confidence interval, WBC = white blood cell, CRP = C-reactive protein.

## Data Availability

The datasets generated during and/or analyzed during the current study are available from the corresponding author on reasonable request.

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
