# Peer review of "Clinical Implication and Prognostic Value of Artificial-Intelligence-Based Results of Chest Radiographs for Assessing Clinical Outcomes of COVID-19 Patients"

_diagnostics, 2023, doi:10.3390/diagnostics13122090_

Round 1

Reviewer 1 Report (Previous Reviewer 4)

The authors have addressed my concerns in the previous round. The revised version can be accepted for publication. 

Author Response

We are grateful for your valuable comments, as they have significantly enhanced the quality of this research.

Reviewer 2 Report (Previous Reviewer 3)

Authors responds to most of my concerns adequately. Few minor issues are listed below.

line 30 - instead of "was aim" use "aimed" or "The aim of the study was"

line 35 - recommend to put a period at "... factors" and start a new sentence "Total O2 ... were evaluated as prognostic indicators of treatment outcome."

line 108 - instead of "they" use "the patient"

line 110 - SpO2 is a prognostic factor of treatment outcome so what does it mean "... prognostic results after the treatment"?

line 185 - Table 1. I recommend that the demographics (sex and age lines) are removed from this table. They are listed in the paragraph above so it is sufficient. This Table is better to focus on the results. The headers should be modified accordingly. In addition, out of curiosity and if I read it correctly, column 4 of this Table indicates that, for Consolidation (%) for example, 463 exams were performed during admission on 136 patients. Some patients had just 1 exam and some had 8, and some in between. Why is this useful to know? Does it indicate something clinically?

Finally, the issue with the equation used for early changes remains (lines 159-162). The logic and justification behind this remains unclear. The authors provided an explanation to justify the results but the only statement they make related to the formula they came up with is that they "proposed an equation to test an idea" (line 312). How and why they came up with it is not clear. So, one cannot decide whether it is the limited sample size (as the authors claim) or the model/equation itself that is a problem.

minor grammar and syntax issues ... see above

Author Response

R2-1. line 30 - instead of "was aim" use "aimed" or "The aim of the study was"

→ Thank you for your comment. We corrected it according to your comment.

R2-2. line 35 - recommend to put a period at "... factors" and start a new sentence "Total O2 ... were evaluated as prognostic indicators of treatment outcome."

→ Thank you for your comment. We made modifications according to your comment.

R2-3. line 108 - instead of "they" use "the patient"

→ Thank you for your comment. We modified it according to your comment.

R2-4. line 110 - SpO2 is a prognostic factor of treatment outcome so what does it mean "... prognostic results after the treatment"?

→ Thank you for your comment. We corrected it accordingly.

R2-5. line 185 - Table 1. I recommend that the demographics (sex and age lines) are removed from this table. They are listed in the paragraph above so it is sufficient. This Table is better to focus on the results. The headers should be modified accordingly. In addition, out of curiosity and if I read it correctly, column 4 of this Table indicates that, for Consolidation (%) for example, 463 exams were performed during admission on 136 patients. Some patients had just 1 exam and some had 8, and some in between. Why is this useful to know? Does it indicate something clinically?

→ Thank you for your comment. In the case of column 4, for example, consolidation, a total of 228 patients underwent 463 exams. Among these, 136 patients had repeated exams, and the number of repetitions ranged from 1 to 8 times. To aid understanding, I moved the number of patients who underwent repeated exams next to the total number of patients in the second column, and in the fourth column, I only included the number of exams. In addition, I changed header accordingly.

The detailed inclusion of the number of repeated measurements is important because this study evaluated not only the initial measurement upon admission but also considered the assessment of treatment response during hospitalization and the overall progression of the patient's disease state. We thought it was important to provide a more comprehensive description of these factors.

R2-6. Finally, the issue with the equation used for early changes remains (lines 159-162). The logic and justification behind this remains unclear. The authors provided an explanation to justify the results but the only statement they make related to the formula they came up with is that they "proposed an equation to test an idea" (line 312). How and why they came up with it is not clear. So, one cannot decide whether it is the limited sample size (as the authors claim) or the model/equation itself that is a problem.

→ Thank you for your comment. We agree with the opinion that there are limitations to the equation. In a clinical setting, as opposed to a prospective study, the timing of the first follow-up CXR after hospitalization can vary among patients. To account for the magnitude of early changes and evaluate treatment response, a mathematical formula was necessary. We consulted with a professional statistician and confirmed that using such a formula to calculate the gradient, slope of numerical changes within the constraints of a retrospective study did not pose significant issues. We have provided additional explanatory details in the methods and discussion sections regarding the approach of calculating the gradient of sequentially measured values. However, if there are concerns about this method, please let us know, and we will consider the option of removing it. We appreciated for your valuable comments. Thank you.

This manuscript is a resubmission of an earlier submission. The following is a list of the peer review reports and author responses from that submission.

Round 1

Reviewer 1 Report

 Hardware implementations using  FPGA  for the algorithms boost performance. 

Author Response

Thank you for your suggestion. The aim of this study was to investigate the significance of using AI software for detecting certain lesions, such as consolidation, in CXRs to evaluate the therapeutic responsiveness of COVID-19 patients. We used a commercially available AI software that had already been developed, and we could not modify its performance since it was not developed by the authors. Because the accuracy of the algorithm in detecting chest lesions was beyond the scope of this study, and due to the limitations of our research, we did not carry out efforts to improve its performance by implementing it on FPGA. However, we used a commercially available AI program with a proven diagnostic performance in previous studies to overcome this limitation [1-5]. Nevertheless, the authors agree that efforts to confirm and upgrade the diagnostic performance of the AI algorithm are important, and we plan to demonstrate this in future research. We have added this limitation to our study. If we did not capture the reviewer's exact point, or if additional modifications are needed, please let us know and we will do our best to meet the requirement.

Reviewer 2 Report

The paper suggests statistical analyses of prognostic value of artificial intelligence(AI)-based result of chest radiographs (CXR) in coronavirus diseases' patients. It explaines all the stages of analyses and finally applies generalized linear mixed model and regression tests to examine the prognastic value of CXR results. From the assessment, it is shown that the selected AI-based software has the same performance of visual diagnosis of CXR in routine practice. 

Considering all the outcomes of the analyses, I consider that there is no  novelty in the statistical analyses part neither the modeling side, nor the general evaluation of the data. Furthermore, as stated by the authors, just a singe AI-based software is used to get the conclusion about the similar performances of softwares and visual diagnosis. Therefore, I think that the paper does not include enough novelties as the article for this journal althought all the explanations are very clear, well organized and detailed. 

Indeed, I see that the paper discussed the performance of AI-softwares for the diagnosis of an important disease and I believe that the authors captures an interesting question for research. Whereas, making such a comment based on a single software can include bias. Because this performance cannot reflect the generality of the AI-based performances. On the other hand, I did not see a new type of modeling or additional analyses which can improve the success of the current analyses too. Therefore, I believe that the presented analyses have been already implemented for other diseases' types as well. Hereby, I conclude that the paper does not have sufficient novelty.

Moreover, I have the following minor comments for the study:

1- Apart from normality check via the Shapiro-Wilk test, there is no control for the homogeneity of the variance for modeling.

2- I did not see the necessity of the beta value in the tables. Could you discuss their interpretations with more details please?

Author Response

Thank you for your comments. We acknowledge the limitations of using a commercial program in this study. However, while many other AI studies present results based on AI algorithms they have developed themselves with their own data, whether these AI algorithms are actually being applied in real clinical environments is another issue. The reason for this is that various barriers are necessary to apply AI algorithms developed for research in actual clinical use, and there is a high possibility that different results may be obtained when compared to research data due to overfitting when applying other datasets or actual clinical settings.

However, in this study, the use of a commercial AI program is a significant difference from other studies, and it is a rare situation that our institution is using the AI program on CXR of all patients in actual clinical practice [1-3]. This study's strength lies in evaluating the practicality and clinical usefulness of using AI in actual clinical practice, beyond diagnosis, but also in treatment response evaluation and prognosis prediction.

Furthermore, even if many hospitals use AI programs in actual patient care as our research team does, it is even rarer to use more than two different AI programs on CXR, and most will selectively use one program. While there is also clinical importance in comparing various AI programs in future research, we believe that this study could serve as a starting point for subsequent studies. Thank you for your comments and if there are any further revisions that the authors need to make, please let us know, and we will try to reflect them as much as possible. Thank you.

R2-1. Apart from normality check via the Shapiro-Wilk test, there is no control for the homogeneity of the variance for modeling.

→ First of all, thank you for your interest in our work, and I think it's this kind of attention to detail, especially the statistical part, that makes our work so valuable. As you may already know, there are many ways to check for homogeneity of variance for modeling. In general, homogeneity of variance refers to the assumption that the variability is the same within a population or group, and in our study, we checked this by 1) visually checking the residuals using residual plots and 2) checking for equality of variance using the Levene’s test. The Levene’s test was chosen because it is less sensitive to the normality of the data. Methods like Bartlett's test are sensitive to the normality of the data, so we decided that Levene's test was more appropriate for our study. For a more general approach. In our study, the results of residual plot and Levene's test confirmed that the equidispersity was satisfied, and the assumptions for applying the regression model were checked, and the test results were reliable.

R2-2. I did not see the necessity of the beta value in the tables. Could you discuss their interpretations with more details please?

→ Thank you for your comments. We noticed that the interpretation of beta values was partially inadequate, and therefore provided a detailed explanation in the first paragraph of our discussion. Additionally, we revised the inconsistent use of the terms 'beta' and 'estimate' in the table to ensure consistency, which was reviewed by a professional statistician among the authors.

We truly appreciate the time and effort taken by the Editor and Reviewers to review our manuscript, and we hope that all of the changes will be found satisfactory.

References

  1. Lee, S.; Shin, H.J.; Kim, S.; Kim, E.K. Successful implementation of an artificial intelligence-based computer-aided detection system for chest radiography in daily clinical practice. Korean journal of radiology 2022,doi:10.3348/kjr.2022.0193.
  2. Kwak, S.H.; Kim, E.K.; Kim, M.H.; Lee, E.H.; Shin, H.J. Incidentally found resectable lung cancer with the usage of artificial intelligence on chest radiographs. PloS one 2023, 18, e0281690.
  3. Shin, H.J.; Lee, S.; Kim, S.; Son, N.H.; Kim, E.K. Hospital-wide survey of clinical experience with artificial intelligence applied to daily chest radiographs. PloS one 2023, 18, e0282123.

Reviewer 3 Report

This paper presents the results of a study on the prognostic value of a commercial AI-based CXR analysis of COVID-19 patients. The study is relatively well designed, the case sample may be small but well selected. Analysis is properly done and results are well justified.  There are few issues (a major one in #3 below) that need to be addressed:

1. Additional information is desirable in the introduction regarding prior AI studies and their prognostic value for COVID 19 patients. References are given but short presentation of major results would be valuable to the reader in order to understand the framework and the value of the present work.

2. The main prognostic features of the study are two scores (consolidation and pleural effusion) provided by the commercial AI. Despite limitations of non-generalizability that are pointed out by the authors, the comparison of a commercial system that is used in clinical practice to the clinical and laboratory outcomes is reasonable and useful to the scientific community.  For those of us, however, unfamiliar with this software, it would be very useful if we had some information on its sensitivity and specificity for consolidation and pleural effusion. In addition, does Csn stand for "consolidation score n???"? Please define. If the AI shows pleural effusion what is the score called?

3. Based on Table 1, the AI's score for pleural effusion is always less than 15%, which is the threshold for detection (line 117). According to the listed value, this score ranges approximately from -7 to +13. What does the negative score mean? Was there ever a marker for pleural effusion from the AI on the CXRs of the study? If not, why did the authors take it into consideration? Does this explain the negative correlation of this feature to the clinical outcome of the  patients? 

4. The parameter used to study early changes and described by the equation in lines 145-147 is very confusing. First, the equation is incomprehensible. Please rewrite in a clearer way. Second, how was this equation derived and what is the justification behind it? It does not seem to hold any prognostic or other value based on the results listed in Table 5 and the reason is unclear. Is it because the way changes are defined? Is it because there is insufficient clinical change? Additional discussion is necessary.

5. Line 135 - Please write Male:Female instead M:F and mean age instead of just mean.

Author Response

R3-1. Additional information is desirable in the introduction regarding prior AI studies and their prognostic value for COVID 19 patients. References are given but short presentation of major results would be valuable to the reader in order to understand the framework and the value of the present work.

→ Thank you for your comments. We have added a brief summary of previous recent studies in the introduction section.

R3-2. The main prognostic features of the study are two scores (consolidation and pleural effusion) provided by the commercial AI. Despite limitations of non-generalizability that are pointed out by the authors, the comparison of a commercial system that is used in clinical practice to the clinical and laboratory outcomes is reasonable and useful to the scientific community.  For those of us, however, unfamiliar with this software, it would be very useful if we had some information on its sensitivity and specificity for consolidation and pleural effusion. In addition, does Csn stand for "consolidation score n???"? Please define. If the AI shows pleural effusion what is the score called?

→ Thank you for your comments. In previous studies, the accuracy for consolidation was 93% (95% CI 91.8-94%) and for pleural effusion it was 99.2% (95% CI 98.8-99.5%) [1]. When considering consolidation, nodule, and pneumothorax together, the AI algorithm's stand-alone performance was an AUC of 0.867 (95% CI 0.858-0.875), with a sensitivity of 88.5% and specificity of 72.3% [2]. We added this in the method section.

In the Figure 1, we added explaination about the AI results. The AI algorithm generates a contour map with abbreviations (where "Csn" stands for consolidation and “Pef” stands for pleural effusion) and its abnormality score for lesion localization, as shown in Figure 1. These abbreviations are pre-defined format in the software. We added this explaination in the method and figure legend sections.

R3-3. Based on Table 1, the AI's score for pleural effusion is always less than 15%, which is the threshold for detection (line 117). According to the listed value, this score ranges approximately from -7 to +13. What does the negative score mean? Was there ever a marker for pleural effusion from the AI on the CXRs of the study? If not, why did the authors take it into consideration? Does this explain the negative correlation of this feature to the clinical outcome of the patients? 

→ Thank you for your comments. The abnormality score represents the probability or likelihood of a lesion based on the AI algorithm's judgment, ranging from 0 to 100%. For the 228 initial CXRs, the score range for pleural effusion was 0.12% to 94.42%, with negative results being impossible by definition. Among them, 220 CXRs had a pleural effusion score of less than 15%, resulting in a higher standard deviation than the mean. We included pleural effusion in the imaging findings because it is known to be accompanied by consolidation in typical pneumonia. However, this result could be seen as reflecting the fact that COVID-19 pneumonia differs from previous bacterial pneumonia in that it is less likely to be accompanied by pleural effusion, while also serving as evidence that AI results reflect differences in imaging findings due to the actual etiology.

R3-4. The parameter used to study early changes and described by the equation in lines 145-147 is very confusing. First, the equation is incomprehensible. Please rewrite in a clearer way. Second, how was this equation derived and what is the justification behind it? It does not seem to hold any prognostic or other value based on the results listed in Table 5 and the reason is unclear. Is it because the way changes are defined? Is it because there is insufficient clinical change? Additional discussion is necessary.

→ Thank you for your comments. While examining patients, we came up with the idea that the pattern of early consolidation changes could be quantified by AI and used to evaluate the patient's treatment response by measuring the degree of increase or decrease. To test this idea, we proposed an equation. However, due to the limitations of retrospective research, it was not possible to obtain consistent results on a uniform date as the follow-up periods for actual patients varied. As an alternative, we devised a method to evaluate the pattern of changes over the first three days of hospitalization. Nevertheless, due to the limited number of patients who underwent follow-up (only 136 out of the total number of patients), most of the predicted outcomes did not yield significant results, as shown in Table 5. However, as the most critical factor was whether the patient's condition improved or worsened, the initial pattern of consolidation response showed significant results, indicating some promise. Therefore, we believe that a well-established prospective large-scale study will be needed to validate this idea in the future. We have revised the description of the equation according to your advice and also added a discussion about it.

R3-5. Line 135 - Please write Male:Female instead M:F and mean age instead of just mean.

→ Thank you. We changed in the method section and Table 1.

We truly appreciate the time and effort taken by the Editor and Reviewers to review our manuscript, and we hope that all of the changes will be found satisfactory.

References

  1. Shin, H.J.; Son, N.H.; Kim, M.J.; Kim, E.K. Diagnostic performance of artificial intelligence approved for adults for the interpretation of pediatric chest radiographs. Scientific reports 2022, 12, 10215.
  2. Jin, K.N.; Kim, E.Y.; Kim, Y.J.; Lee, G.P.; Kim, H.; Oh, S.; Kim, Y.S.; Han, J.H.; Cho, Y.J. Diagnostic effect of artificial intelligence solution for referable thoracic abnormalities on chest radiography: A multicenter respiratory outpatient diagnostic cohort study. European radiology 2022,doi:10.1007/s00330-021-08397-5.

Reviewer 4 Report

This paper investigated the clinical implications and prognostic value of AI-based results for chest radiographs in COVID-19. The results on a cohort of 228 patients show that AI-based CXR results for consolidation have potential prognostic value for predicting treatment outcomes in COVID-19. 

I think this paper is interesting in some aspects such as the importance of AI software in clinical use. However, COVID-19 became less important after a three-year understanding of this disease. My concerns about this paper are listed as follows.

(1) Which variant of COVID-19 dominates the dataset? This could be discussed as it could vary across different variants.

(2) This paper only used one AI-based software available in the hospital. I am wondering what're the advantages of this software over published algorithms such as the one in "An Ensemble Learning Method Based on Ordinal Regression for COVID-19 Diagnosis from Chest CT.

(3) What are the suggestions of the present results for current variants of COVID-19?

Author Response

R4-1. Which variant of COVID-19 dominates the dataset? This could be discussed as it could vary across different variants.

→ Thank you for your comments. This study was conducted between September 2021 and March 2022 on a group of hospitalized patients. From the third week of January 2022, the Omicron variant became predominant in South Korea, while the Delta variant was dominant prior to that period. We have included a mention of this.

R4-2. This paper only used one AI-based software available in the hospital. I am wondering what're the advantages of this software over published algorithms such as the one in "An Ensemble Learning Method Based on Ordinal Regression for COVID-19 Diagnosis from Chest CT.

→ Thank you for your comment with interesting reference. The most significant difference of this program is that it is an AI-based lesion detection software for chest X-ray, which is a basic tool for screening and follow-up and can be easily used for evaluating therapeutic responsiveness of lung infection patients, especially those with diseases such as COVID-19, where the use of chest CT is not readily available. Secondly, while various AI algorithms have been developed by research teams, overfitting has been a major obstacle to their actual clinical application. However, this study used a commercially available program, which makes it possible for other institutions to apply the results. We have added a mention of this in the discussion section.

R4-3. What are the suggestions of the present results for current variants of COVID-19?

→ Thank you for your comments. During the study period, the Delta and Omicron variants were predominant, and since then, the Omicron subvariant has continued to dominate. The differences between the two can be summarized as follows:

Transmission: The Omicron variant is believed to be more transmissible than the Delta variant due to a larger number of mutations, including mutations to the spike protein.

Symptoms: Early reports suggest that the symptoms of the Omicron variant may be milder than those of the Delta variant, although more data is still being collected.

Vaccine effectiveness: There are concerns that the Omicron variant may be more resistant to existing vaccines than the Delta variant.

Clinical outcomes suggest that the proportion of severe cases was higher with Delta than with Omicron infections. However, regardless of the variant, the transition to severe cases can occur equally in elderly patients, patients with underlying diseases, and patients with compromised immune systems. Moreover, no differences in imaging findings have been reported between different variants. Therefore, it is expected that the same effects can be achieved in predicting prognosis and clinically applying lung lesion detection regardless of the variant, as the prediction of prognosis for severe cases is more important than the management of mild cases.

We added this point in the discussion section.

We truly appreciate the time and effort taken by the Editor and Reviewers to review our manuscript, and we hope that all of the changes will be found satisfactory.

Round 2

Reviewer 2 Report

In the current version of the paper, I saw that the authors answered the suggestions  by revising the interpretation of the beta values, application of Levene test for the control of homogeneity and the reason of the single AI-based software in the analyses.

Hereby, I undertstand that the revision and I also read other revisions of the study regarding the anonymous referees' suggestions. But, I regret to say that the paper does not still have enough novelty for a publication of this journal. I read the reason why the authors cannot use more softwares in the analyses, and I am still thinking that the research question is interesting. Whereas, for a journal publication in this type needs to cover more comprehensive analyses even though they can reach single software and majority of the experts in the field uses a single software in their analyses. But currently the conclusion is limted for a sole software although there are alternatines and in order to get a general conclusion about the implementation of AI-based analyses, more softwares need to be considered. Therefore, unfortunately, I think that the analysis part of the study is weak regarding the novelty. 

Reviewer 4 Report

The responses have addressed my previous concerns.